# RobuQ: Pushing DiTs to W1.58A2 via Robust Activation Quantization

## 1. Additional Visualization Results

We provide additional qualitative results from DiT-XL/2 (Peebles & Xie, 2023) at $256\times256$ resolution. Figures 1a and 1b show samples from W1.58A2 and W1.58A4, and Figures 2–11 compare RobuQ (w/o AMP)-A4, RobuQ-A3, and RobuQ-A2, highlighting consistent quality and robustness across activation precisions. All samples are generated with 250 steps (Ho et al., 2020) and cfg (Ho & Salimans, 2022) of 4.0 on ImageNet-1K (Russakovsky et al., 2015) dataset.

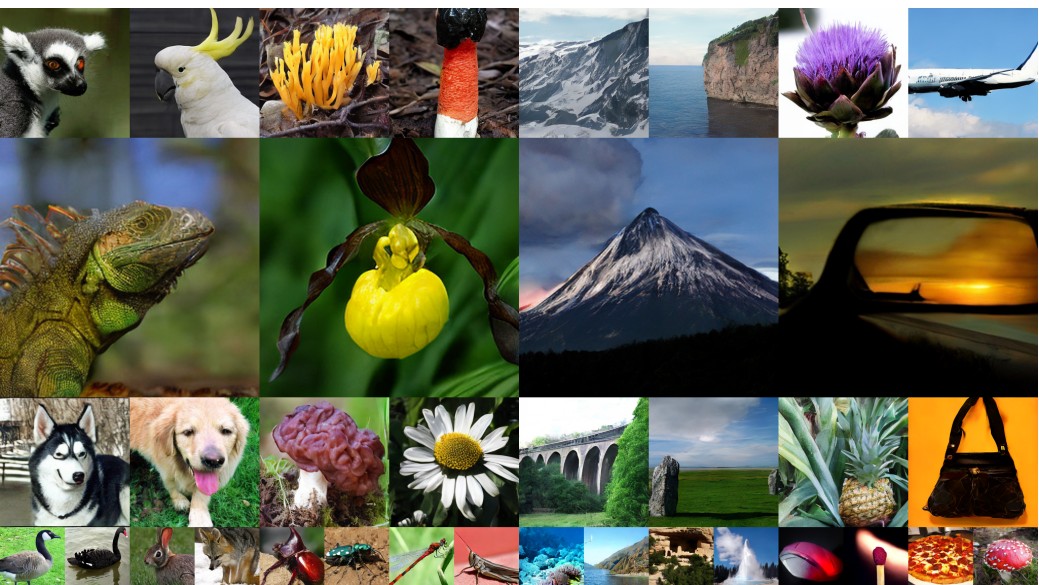

*(a)* RobuQ-A2

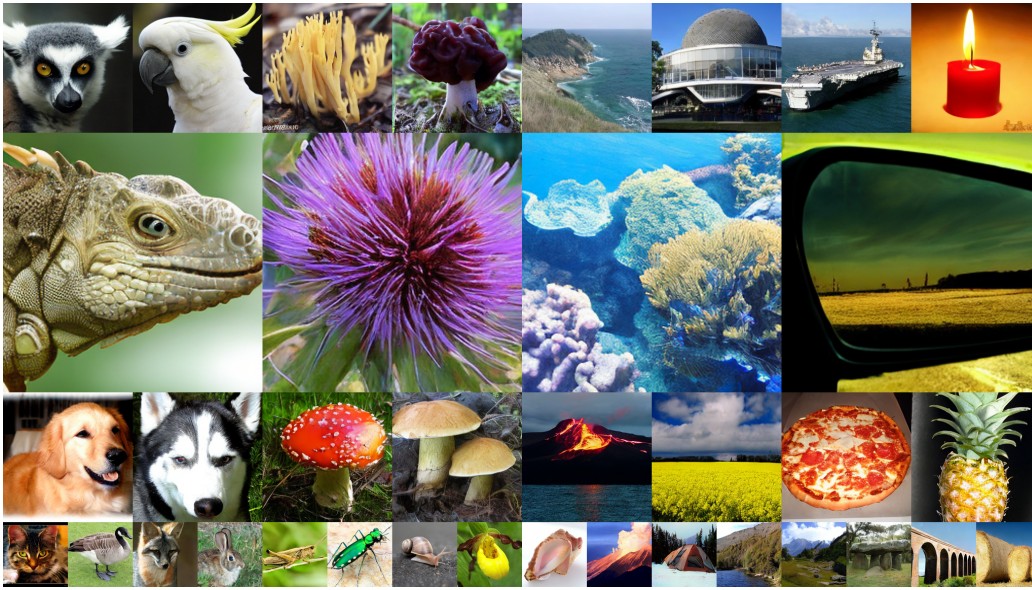

*(b)* RobuQ (w/o AMP)-A4

Figure 1. **RobuQ enables DiTs to generate competitive results at ultra-low bit settings.** We show $256\times256$ samples from W1.58A2 and W1.58A4 quantized models trained on ImageNet-1K.

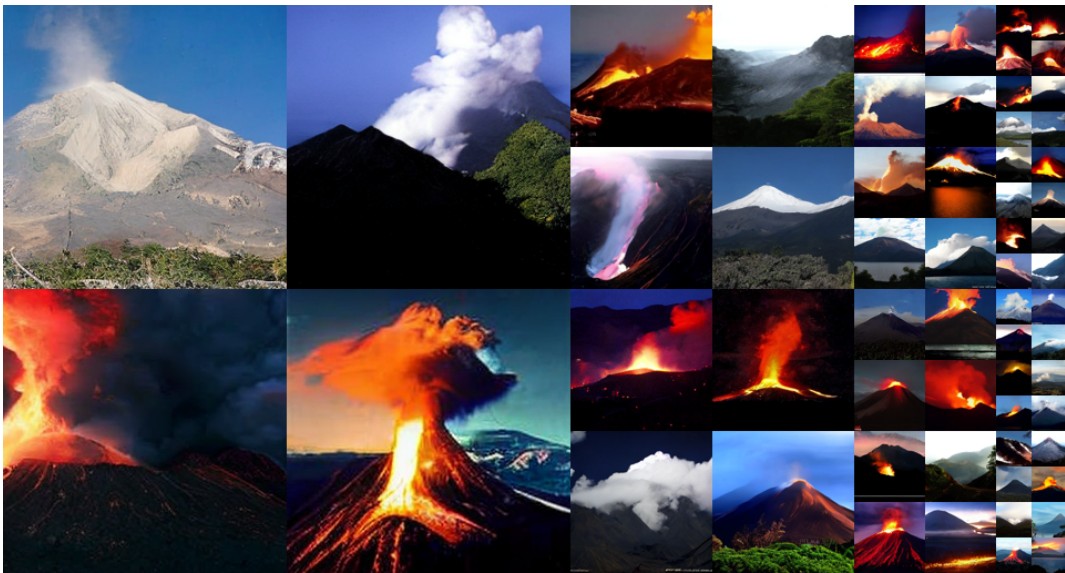

*(a)* RobuQ (w/o AMP)-A4

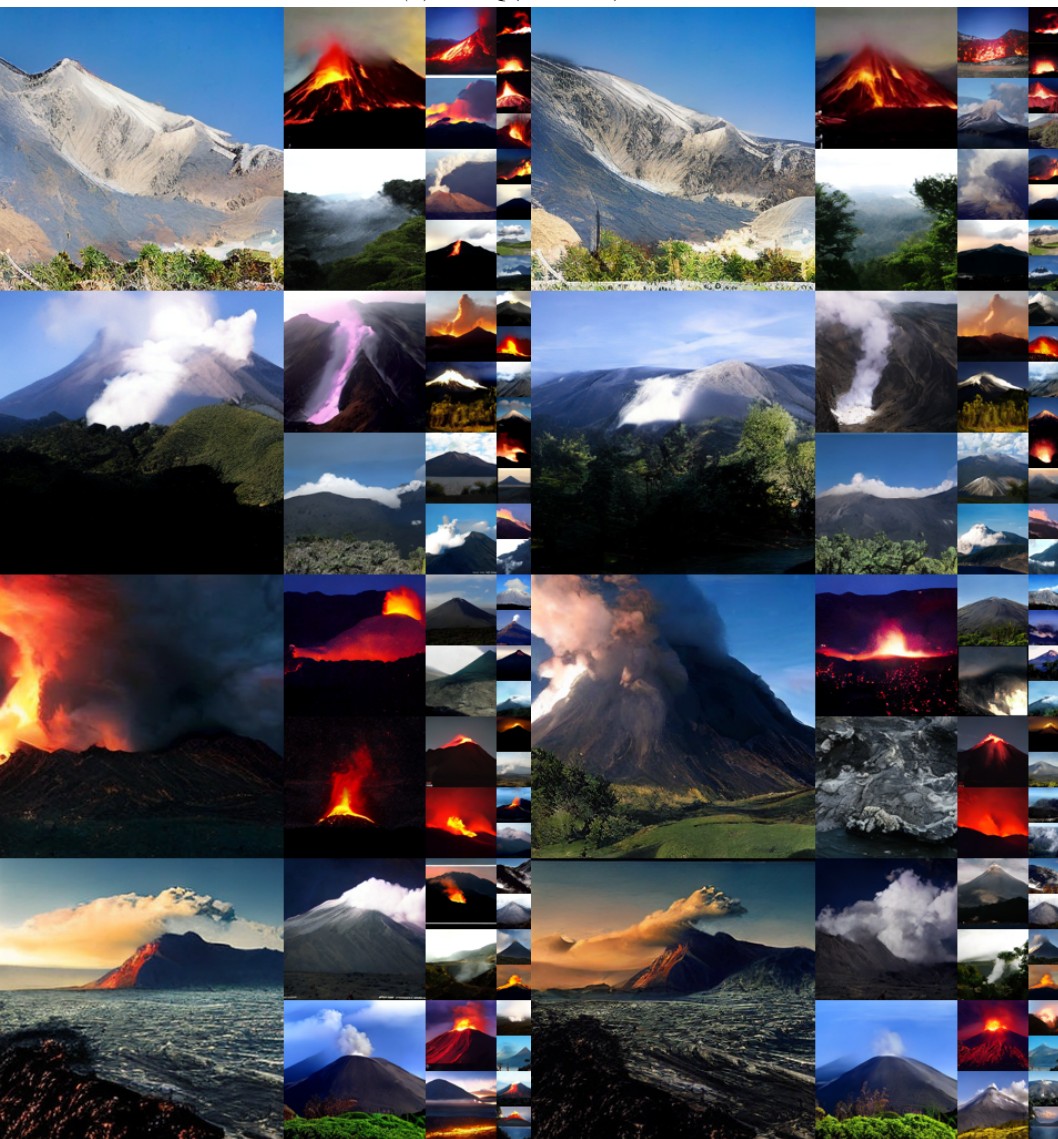

*(b)* RobuQ-A3

*(c)* RobuQ-A2

*Figure 2.* Uncurated DiT-XL/2 256×256 samples, cfg = 4.0, class label = 980

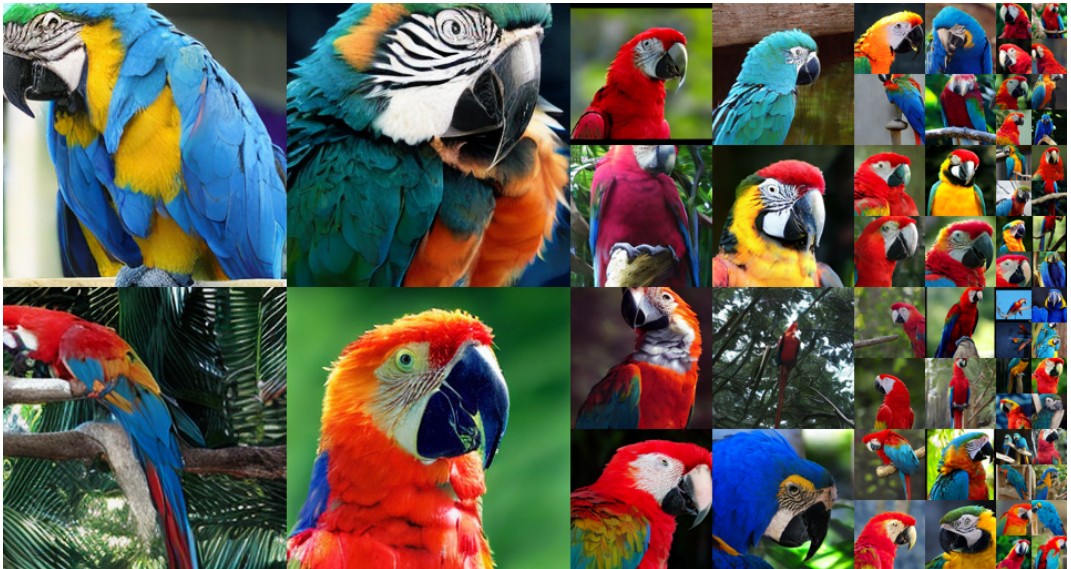

*(a)* RobuQ (w/o AMP)-A4

*(b)* RobuQ-A3      *(c)* RobuQ-A2

*Figure 3.* Uncurated DiT-XL/2 256×256 samples, cfg = 4.0, class label = 88

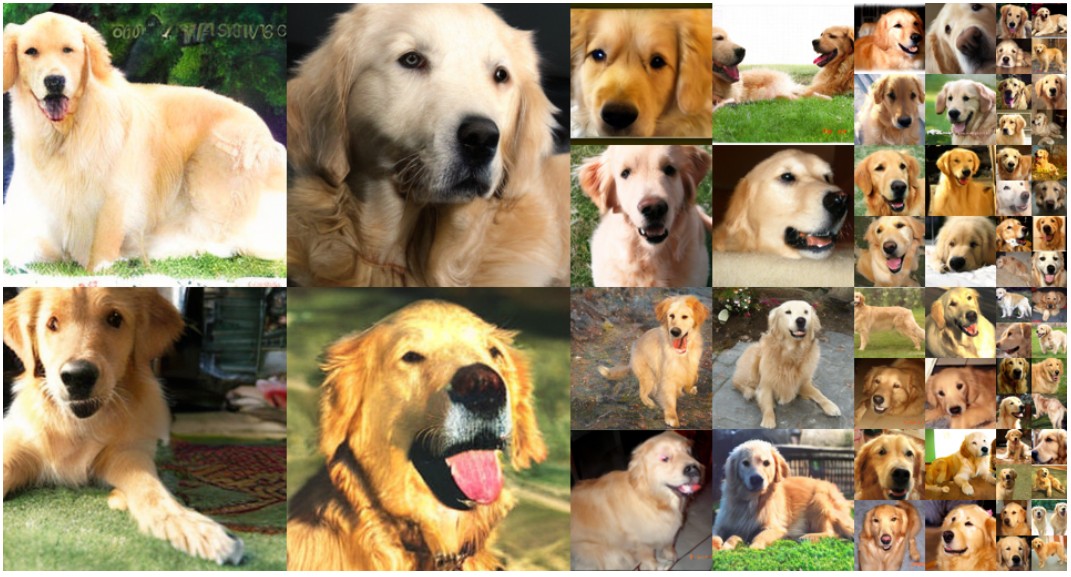

*(a)* RobuQ (w/o AMP)-A4

*(b)* RobuQ-A3

*(c)* RobuQ-A2

*Figure 4.* Uncurated DiT-XL/2 256×256 samples, cfg = 4.0, class label = 207

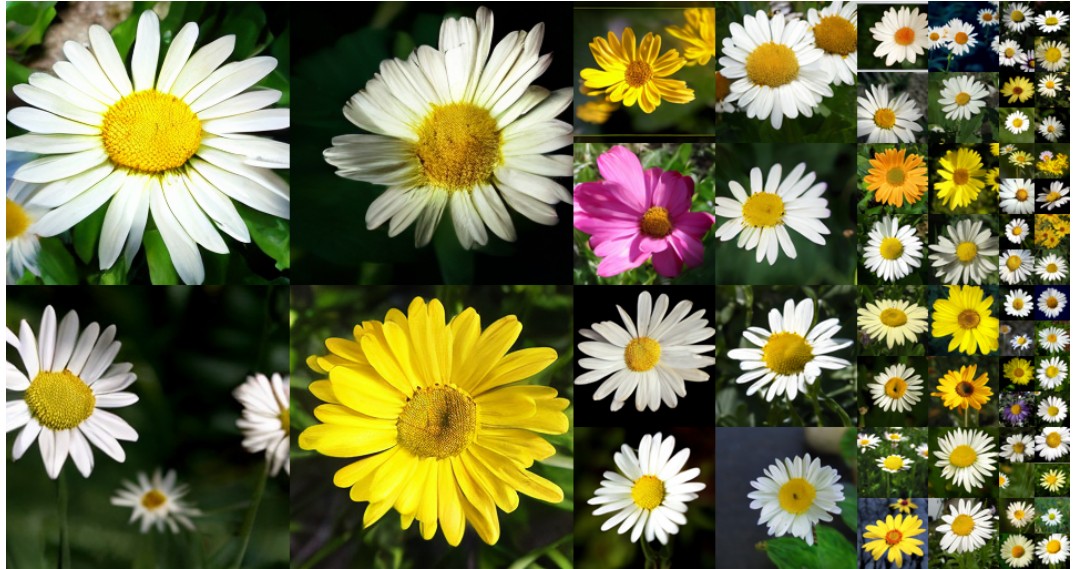

*(a)* RobuQ (w/o AMP)-A4

*(b)* RobuQ-A3

*(c)* RobuQ-A2

5

*Figure 5.* Uncurated DiT-XL/2 256×256 samples, cfg = 4.0, class label = 985

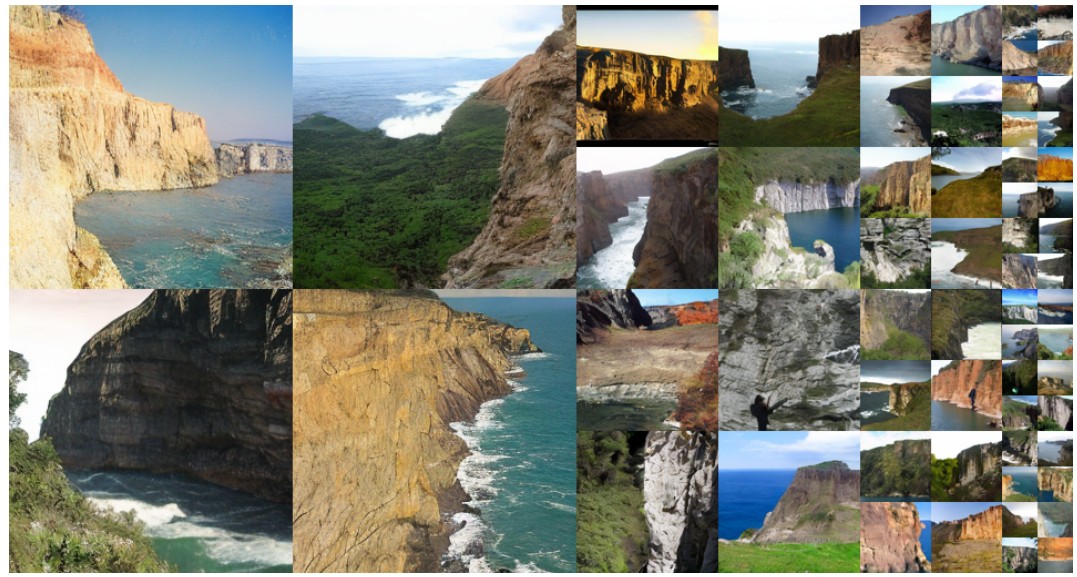

*(a)* RobuQ (w/o AMP)-A4

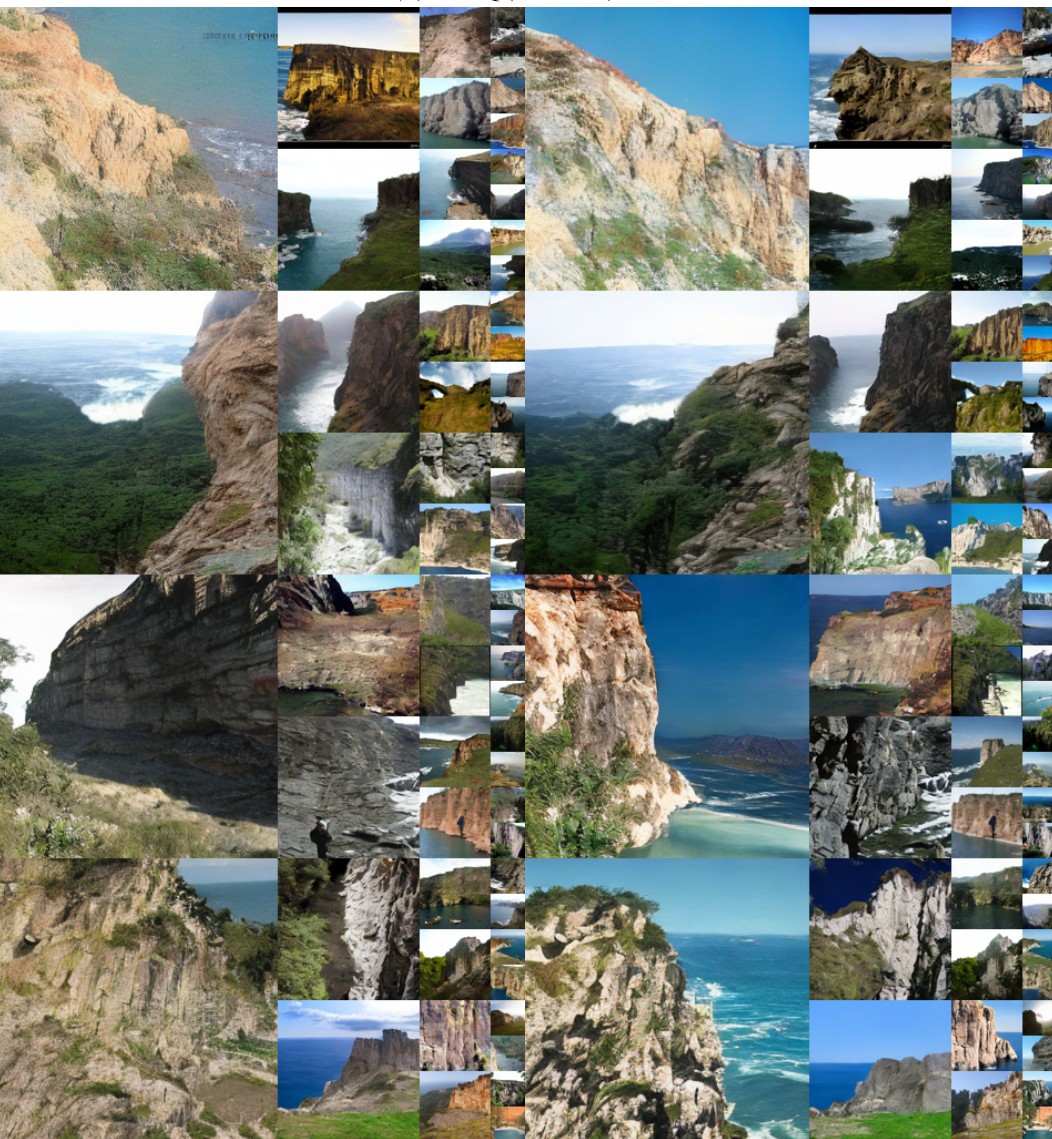

*(b)* RobuQ-A3                 *(c)* RobuQ-A2

*Figure 6.* Uncurated DiT-XL/2 256×256 samples, cfg = 4.0, class label = 972

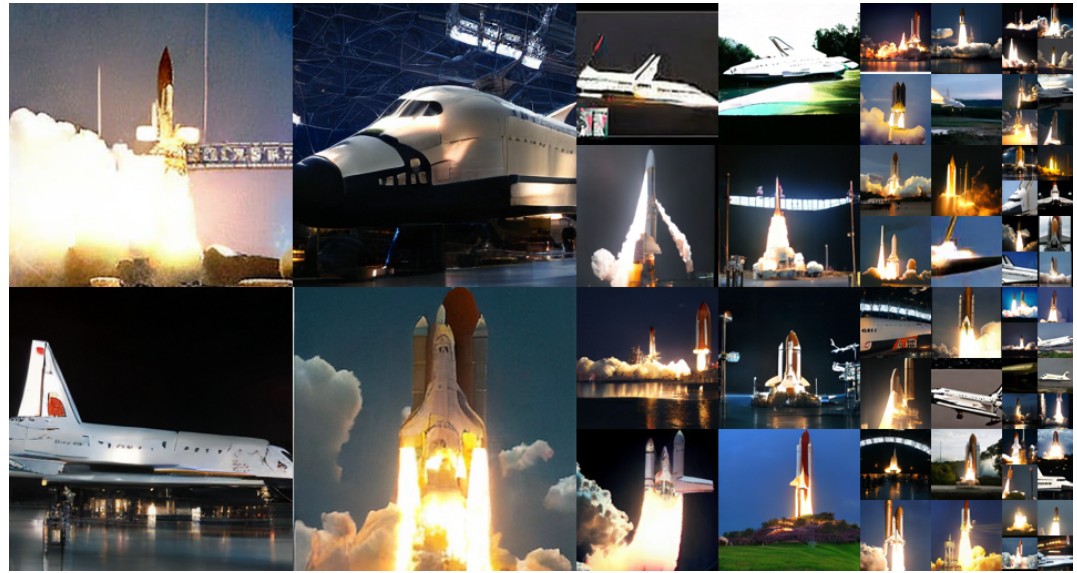

*(a)* RobuQ (w/o AMP)-A4

*(b)* RobuQ-A3                                                    *(c)* RobuQ-A2

*Figure 7.* Uncurated DiT-XL/2 256×256 samples, cfg = 4.0, class label = 812

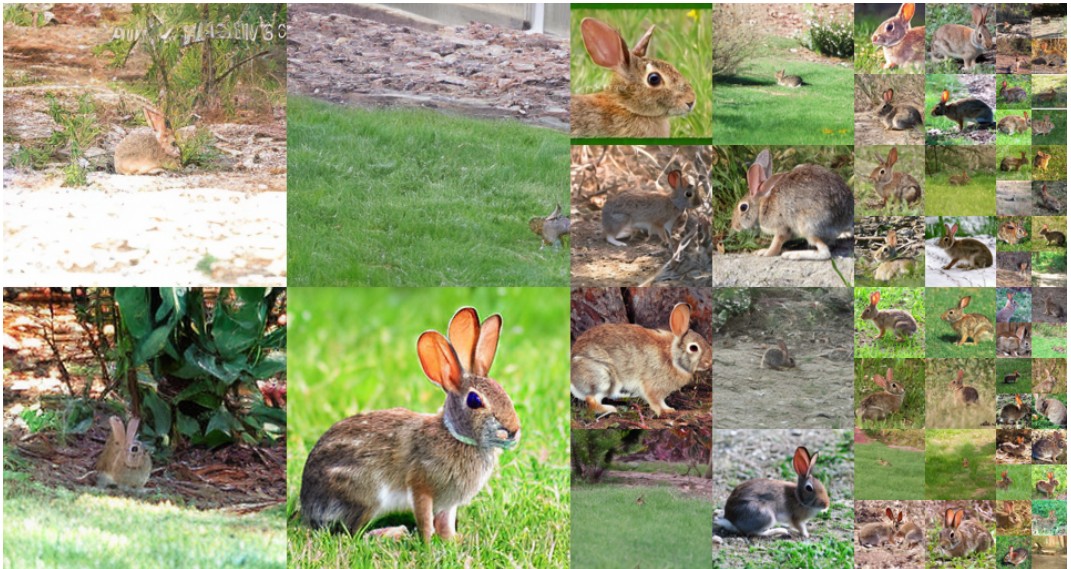

*(a)* RobuQ (w/o AMP)-A4

*(b)* RobuQ-A3          *(c)* RobuQ-A2

*Figure 8.* Uncurated DiT-XL/2 256×256 samples, cfg = 4.0, class label = 330

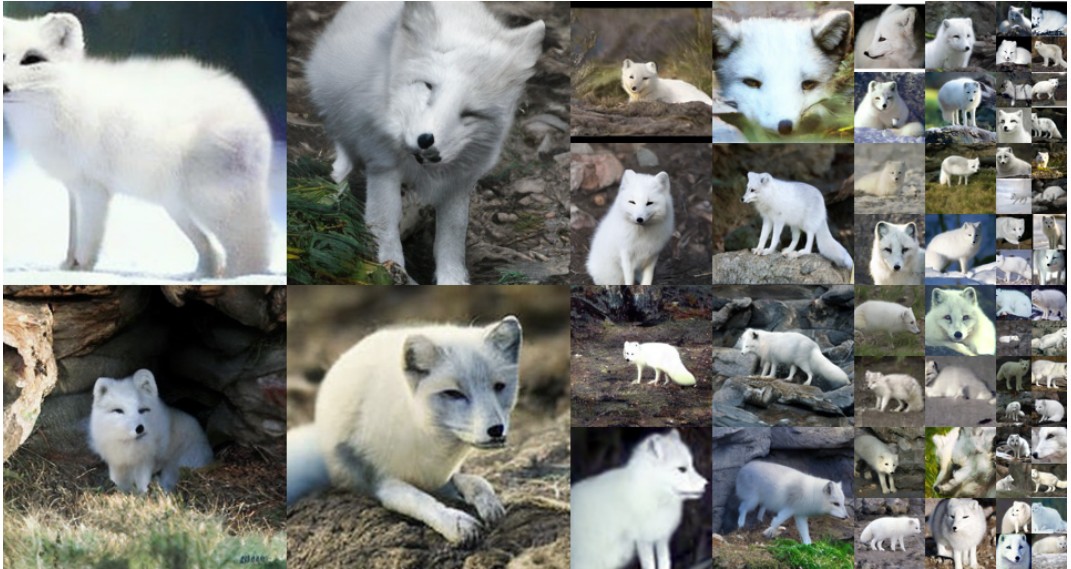

*(a)* RobuQ (w/o AMP)-A4

*(b)* RobuQ-A3        *(c)* RobuQ-A2

*Figure 9.* Uncurated DiT-XL/2 256×256 samples, cfg = 4.0, class label = 279

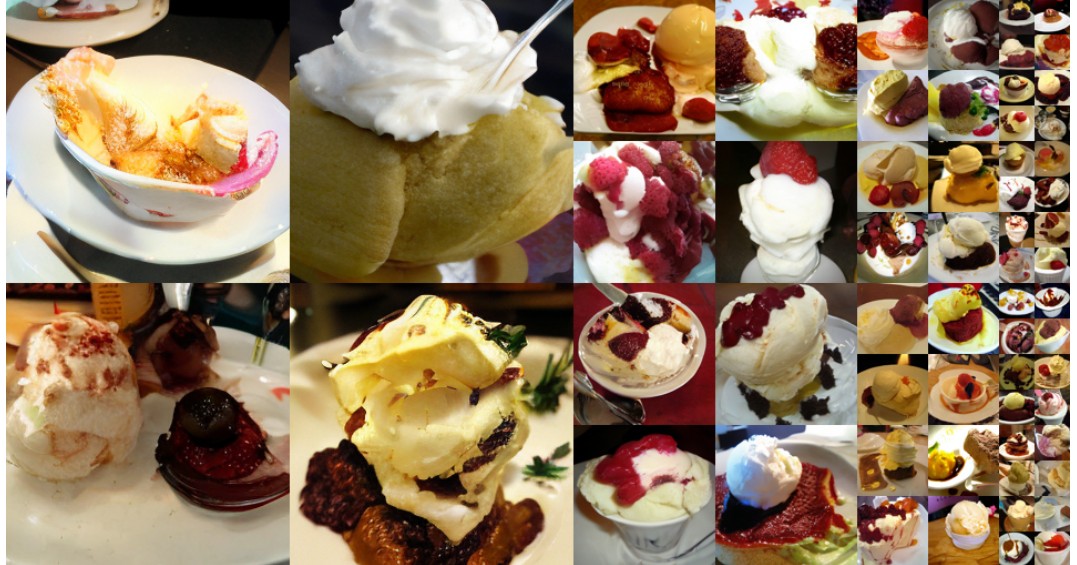

*(a)* RobuQ (w/o AMP)-A4

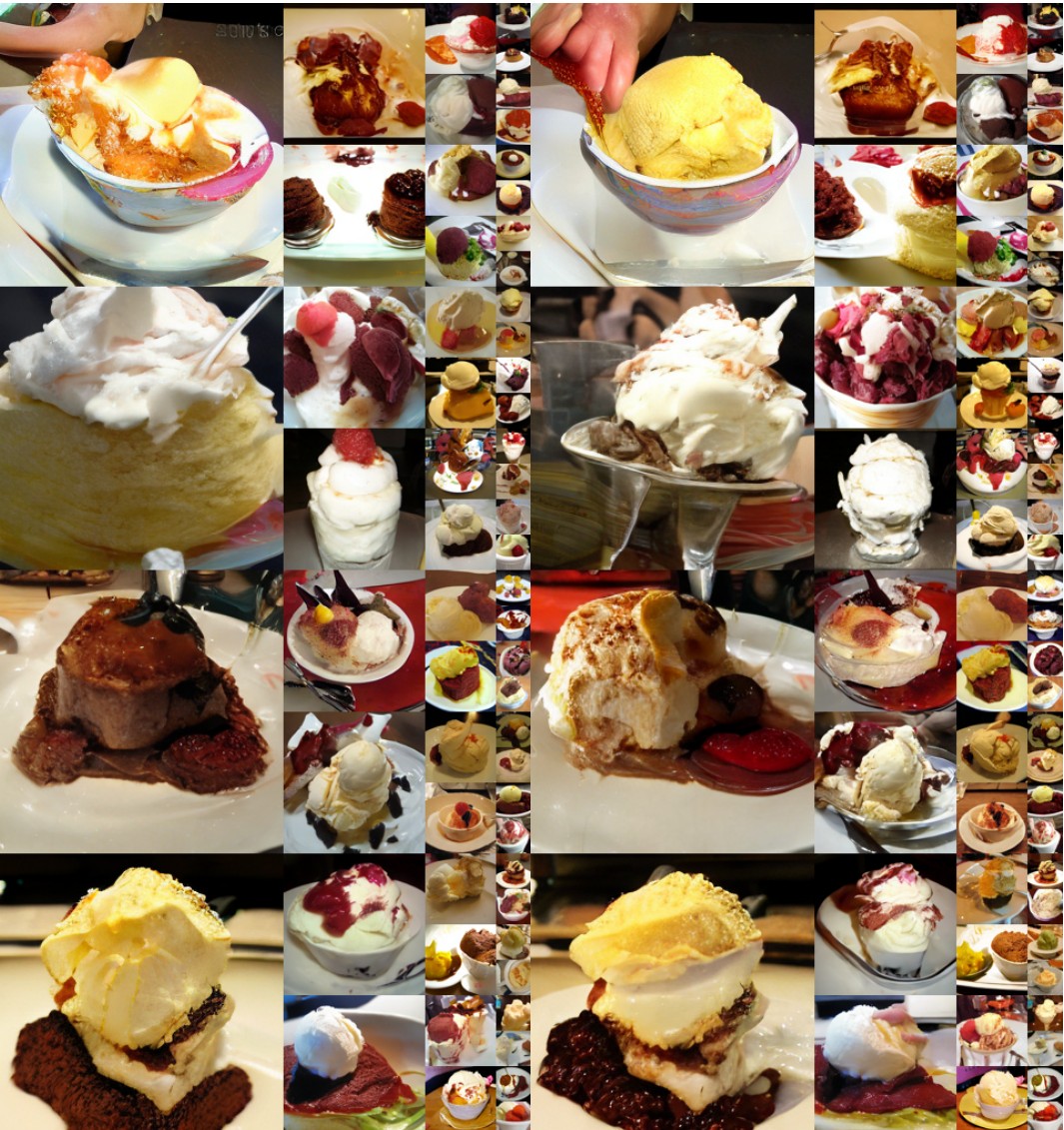

*(b)* RobuQ-A3

*(c)* RobuQ-A2

*Figure 10.* Uncurated DiT-XL/2 256×256 samples, cfg = 4.0, class label = 928

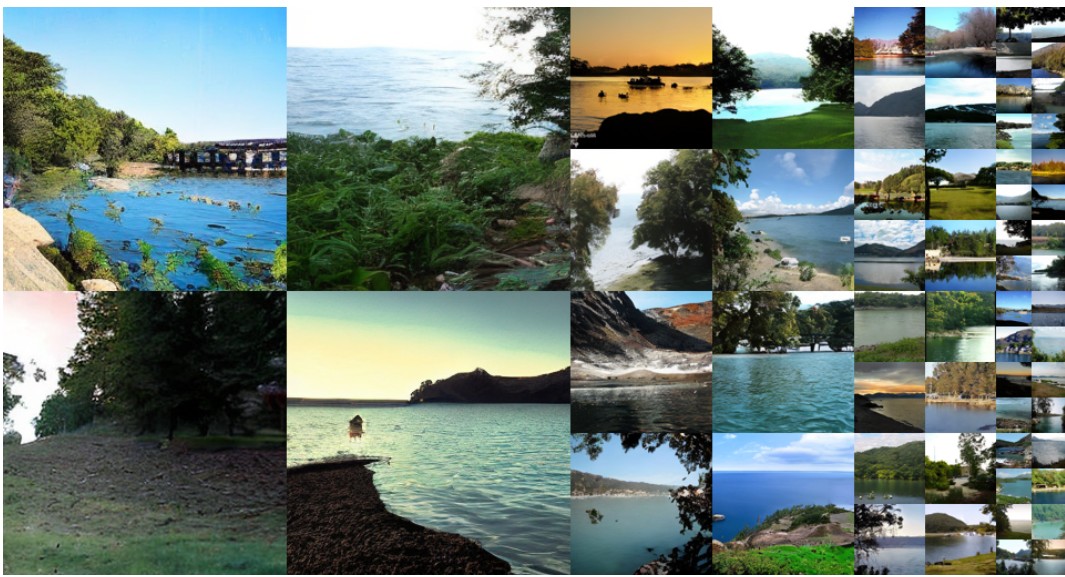

*(a)* RobuQ (w/o AMP)-A4

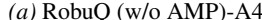

*(b)* RobuQ-A3                                   *(c)* RobuQ-A2

*Figure 11.* Uncurated DiT-XL/2 256×256 samples, cfg = 4.0, class label = 975