# OpenReview forum: "RobuQ: Pushing DiTs to W1.58A2 via Robust Activation Quantization"
_ICML.cc/2026/Conference — ICML 2026 regular_

### Official Review · Reviewer_wukp · 2026-03-04

**Soundness:** 3
**Presentation:** 3
**Significance:** 3
**Originality:** 3
**Overall Recommendation:** 4
**Confidence:** 1

**Summary:**

This paper addresses the challenging problem of ultra-low-bit quantization for Diffusion Transformers (DiTs). The authors identify activation quantization as the primary bottleneck for deploying DiTs in resource-constrained environments. To tackle this, they propose RobuQ, a framework comprising two core components:


RobustQuantizer: Utilizes the Hadamard transform to "Gaussianize" diverse activation distributions, enabling more effective per-token quantization without requiring calibration data.


AMPN (Activation-only Mixed-Precision Network): The first pipeline to focus exclusively on per-layer activation bit-width allocation (while keeping weights ternary), effectively eliminating information bottlenecks.

The framework pushes DiTs to an extreme W1.58A2 (ternary weights, 2-bit average activations) configuration, achieving state-of-the-art results on ImageNet-1K with a 13.2x compression ratio

**Compliance With Llm Reviewing Policy:**

Affirmed.

**Key Questions For Authors:**

1. How does the training overhead of the sensitivity profiling stage in AMPN scale with model size?

2. Could you provide more details on the specific CUDA kernels used to achieve the reported iteration speeds on the RTX 4090?

**Strengths And Weaknesses:**

S1: Successfully quantizing activations to an average of 2 bits while maintaining stable image generation is a notable technical feat.


S2: The theoretical and empirical demonstration that the Hadamard transform can convert irregular DiT activations into a normal distribution provides a solid foundation for the proposed RobustQuantizer.

W1: While the performance on DiT-XL/2 is impressive, the paper primarily focuses on this specific architecture. Further validation on a wider variety of transformer-based generative models (e.g., Video DiTs or larger LLMs) would strengthen the generalizability of the claims.

W2: The AMPN pipeline involves multiple steps, including sensitivity profiling and dynamic programming for bit-width allocation. While effective, this adds complexity to the training process compared to uniform quantization.

---

> ### Author Rebuttal · Authors · 2026-03-31
>
> ## Rebuttal to Reviewer wukp
>
> We thank Reviewer wukp for the positive assessment. Motivated by your feedback, we extended RobuQ to FLUX-Schnell, α-Flow, and MiniMind, and found that the method transfers well across modalities. On MiniMind-MoE, W1.58A3 even slightly outperforms full-precision in perplexity.
>
> > **W3-1: The paper focuses on DiT-XL/2. Further validation on a wider variety of transformer-based generative models will enhance universal applicability.**
>
> A: We added results on three architectures beyond DiT-XL/2:
>
> **(1) FLUX-Schnell** (4-step text-to-image): W4A4 PTQ, FID 18.40→18.57, negligible degradation. **(2) α-Flow** (one/two-step): W4A4 PTQ achieves FID 2.59 at NFE=1 (FP: 2.58) without calibration; W1.58A4 QAT yields 6.01/5.89 for NFE=1/2. See our response to Reviewer JUBs W1-2 & W1-3 for full tables.
>
> **(3) MiniMind** (language model, PPL evaluation) [R8]:
>
> MiniMind-Dense:
>
> | Model | Params | W/A | PPL ↓ |
> |-------|--------|-----|-------|
> | FP | 200M | 32/32 | 6.09 |
> | Naive QAT | 200M | 1.58/32 | 6.67 |
> | RobuQ (Ours) | 200M | 1.58/3 | 6.51 |
> | FP | 64M | 32/32 | 6.4523 |
> | Naive QAT | 64M | 1.58/32 | 7.0058 |
> | **RobuQ (Ours)** | **64M** | **1.58/3** | **6.6622** |
>
> MiniMind-MoE:
>
> | Model | Params | W/A | PPL ↓ |
> |-------|--------|-----|-------|
> | FP | 200M | 32/32 | 5.78 |
> | Naive QAT | 200M | 1.58/32 | 5.84 |
> | RobuQ (Ours) | 200M | 1.58/3 | 5.72 |
>
> Across all settings, RobuQ W1.58A3 outperforms Naive QAT W1.58A32. On 200M MoE, W1.58A3 (PPL 5.72) even slightly surpasses full-precision (5.78). These results, together with FLUX-Schnell, α-Flow, and DiT-XL/2, show that RobuQ transfers across architectures, sampling regimes, and modalities.
>
> > **W3-2: The AMPN pipeline adds complexity to the training process compared to homogeneous quantification.**
>
> A: AMPN adds one extra step (sensitivity profiling), but it is necessary: uniform A2 quantization leads to training collapse (Fig. 5), while AMPN enables stable W1.58A2 generation with IS improving from 200 to 273 (CFG=4.0, 50 steps). The added cost is a one-time offline phase (~9.8h with 8 GPUs); subsequent QAT and inference are unchanged. Moreover, this layer-wise sensitivity analysis can also serve as a tool for probing information bottlenecks in the model, potentially inspiring quantization-aware architecture design.
>
> > **Q3-1: How does the training overhead of the sensitivity profiling phase in AMPN scale with model size?**
>
> A: Let $L$ = number of layers, $S$ = QAT steps per layer during profiling, $T$ = total QAT training steps, $N$ = number of GPUs. The profiling cost is $O(L \times S)$, scaling linearly with layer count. Since each layer is profiled independently, the process is embarrassingly parallel: with $N$ GPUs the wall-clock time reduces to $O(LS/N)$. The main QAT training cost is $O(L \times T)$, so the profiling-to-training ratio is:
>
> $$\frac{O(LS)}{O(LT)} = O\left(\frac{S}{T}\right),$$
>
> which is independent of $L$. This means that as the model grows larger, profiling remains a constant fraction of the total training budget rather than becoming a new scalability bottleneck. For DiT-XL/2 ($L=28$, $S=1000$) on 8 GPUs, this amounts to ~9.8 hours.
>
> > **Q3-2: Can you elaborate on the specific CUDA cores used to achieve the RTX 4090's reporting iteration speed?**
>
> A: Our deployment builds on the Nunchaku [R5] int4 inference engine, with an additional Hadamard fusion strategy integrated into the front-end. The ternary weights are unpacked to int4 representation for execution; the primary benefit of ternary weights at this stage is memory reduction (3914→554 MB). The low-bit linear path consists of two operators:
>
> 1. **Fused front-end**: combines the Hadamard transform, per-token scaling, activation packing, and the LoRA down-projection into a single kernel.
> 2. **Nunchaku int4 GEMM backend**: handles matrix multiplication, dequantization, and the LoRA up-projection.
>
> Online Hadamard is only needed for `o_proj` and `down_proj`; all other projections absorb the transform into weights. For the auxiliary SVD branch, we reuse Nunchaku's branch-parallel execution strategy to overlap it with the main low-bit path. The operator-level breakdown is:
>
> | Shape (`bsz`, `seq`, `in`, `out`) | Front-end (ms) | GEMM backend (ms) | Total (ms) | vs FP32 |
> |---|---:|---:|---:|---:|
> | `(16, 256, 4608, 1152)` | 0.210 | 0.084 | 0.305 | 3.8× |
> | `(16, 1152, 8192, 8192)` | 0.615 | 4.226 | 4.828 | 12.0× |
>
> On smaller DiT-like shapes, the front-end dominates; on larger shapes, the GEMM backend takes over and speedup reaches 12.0×. On FLUX-Schnell v2, inserting online Hadamard changes per-step latency only from 0.2166 s to 0.2202 s (~1.7% overhead), confirming that the overhead is negligible at realistic model scales.
>
> **References:**
>
> [R6] Yuxuan Sun, Ruikang Liu, Haoli Bai, et al. "FlatQuant: Flatness Matters for LLM Quantization." ICML 2025.
>
> [R8] Jingyao Gong. "MiniMind: Train a Tiny LLM from Scratch." GitHub, 2024. https://github.com/jingyaogong/minimind

---

> > ### Author Rebuttal · Reviewer_wukp · 2026-04-05
> >
> > I will keep my original rating.

---

> > > ### Author Response · Authors · 2026-04-06
> > >
> > > We are glad that our rebuttal has fully addressed your concerns.  Should any further questions arise, we are happy to provide additional results or clarifications at any time.

---

### Official Review · Reviewer_k3im · 2026-03-11

**Soundness:** 2
**Presentation:** 3
**Significance:** 2
**Originality:** 3
**Overall Recommendation:** 4
**Confidence:** 3

**Summary:**

The paper proposes RobuQ, an ultra-low-bit quantization framework for Diffusion Transformers (DiTs). The approach combines an SVD-initialized baseline, a Hadamard-based activation transformation, and an Activation-only Mixed-Precision Network (AMPN) to achieve stable image generation at W1.58A2/A3 configurations.

**Compliance With Llm Reviewing Policy:**

Affirmed.

**Final Justification:**

Although the hardware-friendliness and acceleration benefits of the method in real-world deployment are difficult to guarantee, this paper achieves significant performance improvements for DiTs with extremely low-bit quantization. The authors’ rebuttal has addressed most of my concerns, so I am raising my score from 3 to 4.

**Key Questions For Authors:**

1. Table 6 currently lacks actual deployment data for the W1.58A2 and W1.58A3 mixed-precision configurations. Could you please provide the real-world throughput or latency data for the A2 and A3 configurations?

2. Could you provide a more detailed runtime breakdown of the extra online overhead introduced by the Per-token Gauss Quantizer (calculation of means, variances, and standard deviations)?

**Limitations:**

The paper lacks a discussion on methodological limitations, particularly regarding the actual speedup achieved in real-world deployment.

**Strengths And Weaknesses:**

Strengths
1. The authors successfully establish a robust and much-needed baseline for ultra-low-bit Quantization-Aware Training in DiT models.

2. The mathematical analysis is sound and elegant. The utilization of the Hadamard transform to convert unknown, irregular activation distributions into predictable, near-independent Gaussian distributions is innovative and supported by rigorous proofs.

Weaknesses
1. Per-token dynamic activation quantization is hardware-unfriendly. Its dynamic scaling disrupts operator fusion and creates memory bottlenecks, undermining the practical acceleration.

2. The Per-token Gauss Quantizer requires real-time statistical calculations. However, these heavy floating-point costs are omitted from the theoretical FLOPs, making the acceleration claims seem overly optimistic.

3. The QAT-based Sensitivity Profiling for the mixed-precision allocation introduces substantial computational costs. The recommended configuration (1,000 steps) requires 78.5 hours solely for metric collection (Table 2c). This high training cost limits the method's scalability to larger models or higher-resolution diffusion tasks.

4. While achieving W1.58A2, the method introduces expensive online operations (FP SVD branch, online Hadamard, real-time per-token statistics) that bottleneck actual efficiency. Consequently, the 11.3x theoretical FLOP reduction (Table 6) translates to a 3.5x actual speedup.

---

> ### Author Rebuttal · Authors · 2026-03-31
>
> We thank Reviewer k3im for the rigorous review. We will demonstrate that: (1) the modest end-to-end speedup is due to DiT-XL/2's small GEMM shapes, not our method — larger matrices approach the theoretical regime; (2) the Per-token Gauss Quantizer with online Hadamard overhead is small and diminishes as matrix dimensions grow.
>
> > **W2-1, W2-2 & W2-4: The dynamic activation quantization of each token is not hardware-friendly; Per-token Gaussian quantization requires real-time statistics; and the theoretical `11.3x` FLOPs reduction becomes only about `3.5x` speedup in practice.**
>
> A: We agree that weighted FLOPs and measured wall-clock speed should be clearly separated. We address these concerns point by point:
>
> **(1) The Per-token Gauss Quantizer is lightweight.** After Hadamard Gaussianization, we estimate the scale via token-wise mean absolute value:
>
> $$\sigma \approx \sqrt{\frac{\pi}{2}} \cdot \mathbb{E}[|x|], \qquad x \sim \mathcal{N}(0,\sigma^2).$$
>
> This requires only one scalar reduction per token — same as common `absmax` quantization — and can be fused into a single front-end kernel.
>
> **(2) Online Hadamard is only needed for a few layers.** Due to the orthogonality of the Hadamard matrix:
>
> $$\mathrm{MSE}(Q(x)\tilde H,\; x\tilde H) = \|Q(x)-x\|_F^2 = \mathrm{MSE}(Q(x),\; x),$$
>
> the transform can be absorbed into adjacent weights for most layers. Only the non-absorbable `o_proj` and `down_proj` require online Hadamard during deployment.
>
> **(3) End-to-end gap comes from model-level factors.** Our `W1.58A4` unpacks ternary weights to int4 for execution, so throughput should be compared against W4A4. The primary benefit of ternary weights is memory (3914→554 MB, 7.1×). SVDQuant [R5] W4A4 also achieves only ~3.8× on DiT-XL/2, comparable to our 3.52×.
>
> | Setting | Speedup (e2e) | Speedup (linear) | Memory | Note |
> |---|---:|---:|---:|---|
> | RobuQ `W1.58A4` | 3.52× | — | 3914→554 MB | int4 backend |
> | RobuQ w/o Hadamard | 3.82× | — | — | Hadamard ≈ 7.7% overhead |
> | SVDQuant [R5] `W4A4` | ~3.8× | — | — | same-scale baseline |
> | Wide DiT, `hidden=2560` | 4.45× | 6.39× | — | nonlinear = 40.3% of runtime |
> | Wide DiT, `hidden=4096` | 6.12× | 8.74× | — | linear approaches 8× regime |
>
> **(4) Larger matrices approach the theoretical speedup.** (`none` = absorbable, `had` = online Hadamard, `fused` = deployed front-end + GEMM):
>
> | Shape | Speedup: `none` | `had` | `fused` | Breakdown: front-end (ms) | GEMM (ms) | total (ms) |
> |---|---:|---:|---:|---:|---:|---:|
> | `(16,256,4608,1152)` DiT | 5.14× | 3.37× | 3.83× | 0.210 | 0.084 | 0.305 |
> | `(16,1152,8192,8192)` large | 12.03× | 9.29× | 11.97× | 0.615 | 4.226 | 4.828 |
>
> | Model | Setting | Per-step latency | Peak memory | Overhead |
> |---|---|---:|---:|---:|
> | FLUX-Schnell v2 | w/o Hadamard | 0.2166 s | 15.95 GB | — |
> | FLUX-Schnell v2 | w/ Hadamard | 0.2202 s | 15.98 GB | ~1.7% |
>
> On smaller shapes the front-end dominates; on larger shapes the GEMM backend takes over. We additionally measure on FLUX-Schnell v2, whose hidden dimension (3072) and sequence length are much larger than DiT-XL/2 (1152), to verify that the overhead remains negligible at a more practical model scale. The speedup scales with matrix size; the DiT-XL/2 result is a conservative lower bound.
>
> > **W2-3: QAT-based mixed-precision allocation sensitivity profiles present considerable computational costs (78.5 hours). This high training cost limits scalability.**
>
> A: The 78.5h (Table 2c) is on a single GPU. Since profiling is embarrassingly parallel across layers, multi-GPU cost drops significantly:
>
> | GPUs | Profiling time | QAT training time | Profiling / Total |
> |---:|---:|---:|---:|
> | 1 | 78.5 h | 126.0 h | 38.4% |
> | 4 | ~19.6 h | 126.0 h | 13.5% |
> | 8 | ~9.8 h | 126.0 h | 7.2% |
>
> With 8 GPUs profiling takes only ~9.8h. Once determined, the bit-width allocation is fixed with zero overhead during training and inference.
>
>
> > **Q2-1 & W2-5: Could you provide real-world throughput for W1.58A2/A3? The paper lacks a discussion on deployment limitations.**
>
> A: Due to the limited rebuttal period, we were unable to develop a dedicated mixed-precision (e.g., A2/A3) deployment kernel within one week. **Our primary goal in this work is to demonstrate the feasibility of stable generation at W1.58A2, rather than to claim a fully optimized deployment stack.** We note that prior works on mixed-precision activation quantization (e.g., ABQ-LLM) have shown that such mixed-bit schemes are practically meaningful. Moreover, existing ultra-low-bit diffusion quantization works (e.g., BiDM, BinaryDM) also do not provide deployed kernel implementations. We plan to develop efficient mixed-precision kernels as a follow-up. We will move the existing limitation discussion from the Appendix to the main text and supplement it with the hardware analysis from this rebuttal.
>
> **References:**
> [R5] Muyang Li, Yujun Lin, Zhekai Zhang, et al. "SVDQuant: Absorbing Outliers by Low-Rank Components for 4-Bit Diffusion Models." ICLR 2025.

---

> > ### Author Rebuttal · Reviewer_k3im · 2026-04-02
> >
> > Thank you to the authors for the thoughtful responses and additional experiments. Most of my concerns have been addressed. Although the per-token quantizer still lacks validation through a real kernel implementation, the authors provide a theoretically convincing explanation. Therefore, I am willing to increase my score.

---

> > > ### Author Response · Authors · 2026-04-03
> > >
> > > We sincerely thank you for the constructive engagement and for raising the score. We appreciate the acknowledgement that our responses have addressed most concerns.
> > >
> > > Regarding the remaining point on real kernel validation for the per-token quantizer: we would like to highlight that the FLUX-Schnell v2 latency measurements reported in our rebuttal (per-step latency: 0.2166s → 0.2202s, ~1.7% overhead) were obtained using our actual deployed kernel, which fuses the Hadamard transform, per-token scaling, activation packing, and LoRA down-projection into a single CUDA kernel. This confirms that the per-token Gauss Quantizer overhead is minimal in practice. We agree that a dedicated mixed-precision (A2/A3) deployment kernel would further strengthen the empirical validation, and we plan to develop this in a follow-up.
> > >
> > > Thank you again for your thoughtful review and timely response. We welcome any further questions or suggestions you may have.

---

### Official Review · Reviewer_JUBs · 2026-03-23

**Soundness:** 3
**Presentation:** 2
**Significance:** 2
**Originality:** 3
**Overall Recommendation:** 4
**Confidence:** 2

**Summary:**

This paper considers both extreme low-bit rate weight and activation quantization for DIT when generating high-quality images.  The weight quantization is realized by making use of the Hadamard transformation and a truncated SVD for compensation, which are inspired by the design of BitNetv2 and BiMaCoSR.  The activation quantization is realized by applying Hadamard transformation and dynamic programing of bit-allocation across neural layers.  The Hadamard transformation helps to convert arbitrary distribution of activation values to an approximate Gaussian distribution, making it easier to do activation quantization in a unified manner.

**Compliance With Llm Reviewing Policy:**

Affirmed.

**Final Justification:**

The authors have addressed my concerns in the rebuttal phase. It looks like the proposed method works even for models with a few sampling steps despite the fact that it applies a few different tricks in a not elegant manner.  I therefore decide to raise the score.

**Key Questions For Authors:**

(1) The author should explain the clamp operation.

(2) Does the solution for the DP program in (10) and (13) depend on the sampling timesteps when generating 1k examples?

(3) The training time in hours are not reported in the paper. Is it expensive to run DP in the training process? Does the method need to generate the 1k examples from time to time in the training processes?

**Limitations:**

See my comments above.

**Strengths And Weaknesses:**

Strengths:

(1) The paper proposes RobuQ by performing effective weight and activation quantization, which is found to produce SOTA image generation performance at extreme low-bit rates.

(2) Dynamic programing of bit-allocation across neural layers in DiT is proposed to optimally allocate the bit rates.

Weaknesses:
(1) The obtained FIDs for ImageNet in the experimental section are within a high range. Usually the FID for ImageNet without quantization could be pushed to within 10 (typically 3 or 4). It is not clear to me what the main reason is in the paper for not obtaining those low FIDs.  Is it because of limited training iterations or because of the quantization effect in the training processes? If it is because of the limited training iterations, I suggest the authors to increase the training iterations to push the FIDs to be in low range range.

(2) Following (1), nowadays diffusion models with a few sampling steps  (check for example, consistency model) or drifting model with one sampling step could obtain FID in the range between 1.5 and 4 for ImageNet.  Even if there are no quantization, the sampling time of those models is very effective. On the other hand, this paper considers sampling steps of from 50 to 150 and obtain FID of more than 10.

(3) If the authors could show that their quantization approach can be effectively applied to those diffusion models of very limited sampling steps and still achieve reasonable FIDs, it will be very beneficial to industry.

---

> ### Author Rebuttal · Authors · 2026-03-31
>
> ## Rebuttal to Reviewer JUBs
>
> We thank Reviewer JUBs for the thoughtful review. Inspired by your suggestion, we conducted PTQ and QAT experiments on one-step generation models as well as the standard 50k-sample protocol, achieving **FID < 10 and even < 3** in several settings. Details below.
>
> > **W1-1: The obtained FIDs for ImageNet are within a high range (>10). Is it because of limited training iterations or the quantization effect?**
>
> A: The seemingly high FID in Table 1 is because we report FID-10k, following prior DiT quantization works (PTQ4DiT [R1], Q-DiT [R2]) for fair comparison. We directly evaluate the public pretrained DiT-XL/2 checkpoint. For completeness, we additionally report 50k-sample results:
>
> | Method | CFG | W/A | FID ↓ | sFID ↓ | IS ↑ |
> |--------|-----|-----|-------|--------|------|
> | FP | 1.5 | 32/32 | 3.71 | 7.85 | 242.21 |
> | Ours (RobuQ) | 1.5 | 1.58/4 | 18.18 | 15.37 | 95.90 |
> | Ours (RobuQ) | 1.5 | 1.58/2 | 27.82 | 19.07 | 66.81 |
> | Ours (RobuQ) | 4.0 | 1.58/4 | 10.68 | 8.18 | 351.29 |
> | Ours (RobuQ) | 4.0 | 1.58/2 | 8.51 | 11.00 | 266.77 |
>
> Under 50k protocol, the full-precision model reaches FID = 3.71 (CFG = 1.5), in the expected range. More importantly, low-bit models benefit strongly from higher CFG: W1.58A4 improves from 18.18 to 10.68, and W1.58A2 from 27.82 to 8.51 (< 10) — already surpassing early one step model like Shortcut-XL/2 [R4] (FID = 10.6 in 1 step). We will distinguish FID-10k from FID-50k and include these results in the revision.
>
> > **W1-2 & W1-3: Few-step diffusion models (e.g., consistency model) obtain FID 1.5–4. Can the quantization approach be applied to those models?**
>
> A: We conducted experiments on two representative fast-generation models.
>
> First, W4A4 PTQ on FLUX-Schnell [R7] (4-step, evaluated on MJHQ 5k samples):
>
> | Method | W/A | FID ↓ | CLIPIQA ↑ | CLIPScore ↑ |
> |--------|-----|-------|-----------|-------------|
> | FLUX-Schnell (FP) | 32/32 | 18.40 | 0.9399 | 26.54 |
> | FLUX-Schnell + Ours (PTQ) | 4/4 | 18.57 | 0.9266 | 26.34 |
>
> Degradation is negligible (FID: 18.40 → 18.57).
>
> Second, we extend RobuQ to α-Flow [R3], a one/two-step generative model (we use α-Flow rather than MeanFlow because the latter's JVP dependency conflicts with our Fast Hadamard Transform path):
>
> | Method | Size (MB) ↓ | W/A | Type | FLOPs (G) ↓ | FID ↓ (NFE=1) | FID ↓ (NFE=2) |
> |--------|-----------|-----|------|-----------|---------------|---------------|
> | α-Flow-XL/2 | 2704.0 | 32/32 | FP | 119.4 | 2.58 | 2.15 |
> | α-Flow-XL/2 + Ours | 388.84 | 4/4 | PTQ | 16.97 | 2.59 | 2.23 |
> | α-Flow-XL/2 + Ours | 194.75 | 1.58/4 | QAT | 10.07 | 6.01 | 5.89 |
> | α-Flow-B/2 | 524.0 | 32/32 | FP | 23.01 | 5.40 | 5.01 |
>
> W4A4 PTQ on α-Flow-XL/2 achieves FID 2.59 at NFE=1, nearly identical to the full-precision 2.58 — without any calibration. Even W1.58A4 QAT yields stable results (6.01 / 5.89 for NFE=1/2). These results suggest that quantization and few-step sampling are largely orthogonal: one reduces cost per step, the other reduces the number of steps. We will include these results in the revision.
>
> > **Q1-1: The author should explain the clamp operation.**
>
> A: Sorry that this was not explained clearly. We use
>
> $$\text{clamp}(v,a,b)=\min(\max(v,a),b).$$
>
> > **Q1-2: Does the solution for the DP program in (10) and (13) depend on the sampling timesteps when generating 1k examples?**
>
> A: No. The 1k validation examples used for sensitivity profiling are **randomly sampled across timesteps**. The resulting layer-wise loss gap $\Delta L_\ell(b_\ell)$ is aggregated over these random timesteps, and the DP solution yields a **fixed bit-width allocation** that does not depend on any specific sampling timestep during inference.
>
> > **Q1-3: The training time in hours are not reported. Is it expensive to run DP in the training process? Does the method need to generate the 1k examples from time to time in the training processes?**
>
> A: See Table 2c. Training has two phases: (1) sensitivity profiling (one-time, offline): 78.5h on 1 GPU, or ~9.8h with 8 GPUs since it is embarrassingly parallel across layers; (2) actual QAT: 126.0h on 1 GPU, standard process with no DP or example generation. The DP solver runs in milliseconds; the 1k examples are generated only once during profiling.
>
> **References:**
>
> [R1] Junyi Wu, Haoxuan Wang, Yuzhang Shang, Mubarak Shah, Yan Yan. "PTQ4DiT: Post-training Quantization for Diffusion Transformers." NeurIPS 2024.
>
> [R2] Lei Chen, Yuan Meng, Chen Tang, Xinzhu Ma, Jingyan Jiang, Xin Wang, Zhi Wang, Wenwu Zhu. "Q-DiT: Accurate Post-Training Quantization for Diffusion Transformers." CVPR 2025.
>
> [R3] Huijie Zhang, Aliaksandr Siarohin, Willi Menapace, et al. "AlphaFlow: Understanding and Improving MeanFlow Models." ICLR 2026.
>
> [R4] Kevin Frans, Danijar Hafner, Sergey Levine, Pieter Abbeel. "One Step Diffusion via Shortcut Models." ICLR 2025 oral.
>
> [R7] Black-Forest-Labs. Flux.1, 2024. URL https://blackforestlabs.ai/

---

> > ### Author Rebuttal · Reviewer_JUBs · 2026-04-03
> >
> > I have raised my score accordingly. Will the authors release their code later on?

---

> > > ### Author Response · Authors · 2026-04-03
> > >
> > > We sincerely thank Reviewer JUBs for the positive re-evaluation and for raising the score. We are glad that our rebuttal has fully addressed your concerns.
> > >
> > > Regarding the code release: yes, we will release the full codebase, including training scripts and checkpoints.
> > >
> > > Thank you again for your constructive review and engagement during the discussion period.

---

### Decision · Program_Chairs · 2026-04-30

**Decision:**

Accept (regular)

**Comment:**

The reviewers overall agree this paper is a good contribution to ultra-low-bit quantization for diffusion transformers. There were a few concerns, e.g. the initially high reported FID values, the applicability to few-step models, and validation beyond DiT-XL/2. The rebuttal addressed most of these issues. After rebuttal, reviewers reached a positive consensus, and most researchers noted that their concerns were fully resolved. Therefore, I recommend acceptance of this paper.